# Spin Polarization of Mn Could Enhance Grain Boundary Sliding in Mg

**DOI:** 10.3390/ma15103483

**Published:** 2022-05-12

**Authors:** Vei Wang, Jun-Ping Du, Hidetoshi Somekawa, Shigenobu Ogata, Wen Tong Geng

**Affiliations:** 1State Key Laboratory of Marine Resource Utilization in South China Sea, School of Materials Science and Engineering, Hainan University, Haikou 570228, China; wangvei@xaut.edu.cn; 2Department of Applied Physics, Xi’an University of Technology, Xi’an 710054, China; 3Department of Mechanical Science and Bioengineering, Osaka University, Osaka 560-8531, Japan; jpdu@tsme.me.es.osaka-u.ac.jp; 4National Institute for Materials Science, Tsukuba 305-0047, Japan; 5Center for Elements Strategy Initiative for Structural Materials, Kyoto University, Kyoto 606-8501, Japan

**Keywords:** magnesium alloys, first-principles calculations, grain boundary segregation, grain boundary sliding, bonding charge delocalization

## Abstract

Segregation of rare earth alloying elements are known to segregate to grain boundaries in Mg and suppress grain boundary sliding via strong chemical bonds. Segregation of Mn, however, has recently been found to enhance grain boundary sliding in Mg, thereby boosting its ductility. Taking the Mg ($\bar{2}$114) twin boundary as an example, we performed a first-principles comparative study on the segregation and chemical bonding of Y, Zn, and Mn at this boundary. We found that both Y-4*d* and Mn-3*d* states hybridized with the Mg-3*sp* states, while Zn–Mg bonding was characterized by charge transfer only. Strong spin-polarization of Mn pushed the up-spin 3*d* states down, leading to less anisotropic Mn–Mg bonds with more delocalized charge distribution at the twin boundary, and thus promotes grain boundary plasticity, e.g., grain boundary sliding.

## 1. Introduction

Being the lightest makes Mg and its alloys the most important and interesting metallic materials in both industry and fundamental research. Grain refinement in Mg and its alloys is known to enhance strength due to the large slope in the Hall–Petch relation [1] and ductility due to the activity of grain boundary plasticity such as the non-basal dislocation slip [2] and grain boundary sliding [3]. Grain boundary segregation has a strong impact on the deformation behavior and mechanical properties of the metallic materials. For instance, segregation of B at grain boundaries in Ni strengthens, H and P embrittles grain boundary cohesion [4], while S enhances grain boundary sliding at elevated temperature, and displays an excellent ductility [5]. Many alloying elements have been observed to segregate to the grain boundary in wrought processed Mg alloys [6,7,8,9,10,11,12,13], affecting in different ways the deformation behavior as well as other mechanical properties [14,15]. For example, the solute elements with an atomic radius larger than Mg are prone to embrittle the grain boundary, causing an intergranular fracture similarly to how Ca does in the Mg-Ca binary alloy [10]. Segregation of rare-earth elements, on the other hand, is not only unique in weakening texture [7,8,16,17,18], but also in making grain boundaries the nucleation site for the non-basal dislocation slip [5] and the driving force to suppress grain boundary sliding [15,19].

In contrast, recent papers have reported that segregation of Mn at grain boundaries affects the major plastic deformation mechanism by enhancing grain boundary sliding [19]. The fine-grained Mg–Mn alloy shows a huge elongation-to-failure of more than 100% even at room temperature [20]. The diversity of segregation effect is closely related to the change of electronic structure induced by alloying elements at grain boundaries. One of the aspects of the electronic structure at the segregated grain boundary, bonding strength, has been systematically studied by first-principles calculations [21,22,23]. Bonding strength across the grain boundary is ascribed as the dominant force to resist crack propagation along the boundary. As for the influence of segregated solute elements on grain boundary sliding, several studies on metallic materials other than Mg have been reported [13,24,25,26,27,28]. However, to the best of our knowledge, there are not any reports on such influence in Mg and it remains largely unclear from the perspective of the electronic structure. In this study, we have selected three specific solute elements of Y, Zn, and Mn, with different filling of *d* electrons. We have investigated the effect of solute elements on the localization of bonding electrons in the interlayer plane at a ($\bar{2}$114) twin boundary in Mg and tried to relate it to grain boundary sliding. Since spin polarization has been found to play key roles in stacking and morphology in novel nanostructured materials [29,30], we have paid particular attention to this effect in this study.

## 2. Model and Computation

The grain boundary we chose is the ($\bar{2}$114) twin boundary, which has been shown to have high energy of formation in previous molecular dynamics simulations [31]. We have modeled this grain boundary using a supercell illustrated in Figure 1. This supercell is composed of two identical grains (each contains 16 atomic layers and 64 atoms), which form a twin boundary in between. Atomic layers are labeled as their serial number counted from the twin boundary, and the interstitial position at the boundary is denoted as GB0. The solution energy of an alloying element *X* in a substitutional position, $\Delta H_{s}\left(X\right)$ can be obtained via
(1)$$\Delta H_{s}\left(X\right)=E\left({\mathrm{Mg}}_{127+X}\right)+E\left(\mathrm{Mg}\right)-E\left({\mathrm{Mg}}_{128}\right)-E\left(X\right)$$
where $E\left({\mathrm{Mg}}_{127+X}\right)$ and $E\left({\mathrm{Mg}}_{128}\right)$ are the total energy of the supercell with and without the alloying atom, E(Mg) and E(X) are the total energy of one Mg or *X* atom in its elemental crystal. Likewise, the solution energy of an alloying element *X* in an interstitial position is
(2)$$\Delta H_{s}\left(X\right)=E\left({\mathrm{Mg}}_{128+X}\right)-E\left({\mathrm{Mg}}_{128}\right)-E\left(X\right)$$

The segregation energy of an alloying element is defined as the change in solution energy upon moving from inside the grain, position GB9, to the vicinity of the twin boundary, position GB0, GB1, GB2, or GB3, which has the lowest energy for it.
(3)$$\Delta E_{seg}\left(X\right)=E\left(X@\mathrm{GB}n\right)-E\left(X@\mathrm{GB}9\right)\;\;\;\;n=0,\;1,\;2,\;or\;3$$

A negative value of $\Delta E_{seg}\left(X\right)$ means that X tends to segregate to the grain boundary.

We performed the first-principles density functional theory (DFT) calculations using Vienna Ab initio Simulation Package [32]. The electron-ion interaction was described using the projector augmented wave (PAW) method [33]. The exchange correlation between electrons was treated both with generalized gradient approximation (GGA) in the Perdew–Burke–Ernzerhof (PBE) form [34]. We used an energy cutoff of 300 eV for the plane wave basis set for all systems, both with and without alloying elements, to ensure equal footing. The Brillouin-zone integration was performed within Monkhorst–Pack scheme [35] using a *k* mesh of (3 × 4 × 1) for structural optimization and one of (6 × 8 × 1) for density of states determination. Both the shape of the super cell and the internal coordinates of atoms are optimized. The structural relaxation is continued until the changes of the total energy of the supercell and forces on all the atoms are converged to less than 10^−4^ eV/cell and 2 × 10^−2^ eV Å^−1^, respectively. We generated the preprocessing initial atomic structure and the post-processing electronic structure of the supercells using the VASPKIT code [36].

## 3. Results and Discussion

Our first-principles calculations showed that all these three elements will segregate to the ($\bar{2}$114) twin boundary in Mg, in agreement to experimental observations on Y [6,7,9,15], Mn [19], and Zn [15]. Y segregated to the GB3 position, while both Zn and Mn segregated to the interstitial position GB0 in the boundary layer as they had a much smaller atomic size than Mg. The segregation energy was −0.55, −0.53, and −0.28 eV for Y, Zn, and Mn, respectively. Strong Y–Mg bonding distorted the local atomic structure, and pushed the Y atom to a position in between the GB2 and GB3 layers. To examine the bonding characters of the segregants, we plot the calculated charge redistribution in each supercell upon formation of the X–Mg bond (X = Y, Zn, Mg, and Mn) in Figure 2. It was obtained for each system by subtracting the superimposed charge density of a free atom X and a rigid supercell with missing X from the charge density of the corresponding optimized supercell. The information for the Mg–Mg bond in a clean system serves as a reference. Figure 2a displays the position of the two planes in which we display the charge-density redistribution. Shown in Figure 2b is the charge density redistribution in the (0$\bar{1}$10) plane passing through the X atom, together with the atoms in GB1 and upper GB2 layers, and in Figure 2c is shown the ($\bar{2}$114) plane, positioned in the middle of GB1 and GB2 layers.

It is clearly seen in Figure 2b that Y lost electrons to Mg, while both Zn and Mn gained electrons from Mg. This is readily understandable if we recall that Y has an electronegativity (1.22) lower than Mg (1.31), whereas Zn (1.65) and Mn (1.55) are more electronegative than the matrix atoms [37]. From the outer circular region of charge depletion, we can judge that Mn gained more electrons from the surrounding Mg than does Zn. This is an example that electronegativity comparison is inaccurate for prediction of charge transfer. We note that the charge accumulation in the core region of Y had a larger extension than Mn, due to the more expansive distribution of Y–4*d* states than Mn–3*d* states. We read out the calculated minimum and maximum of charge density difference (in *e*/Å^3^) in both panels and list them in Table 1. It was found that in plane ($\bar{2}$114), the GB1–GB2 interlayer, that the magnitude of spatial variation of binding strength had the order of Y > Zn > Mg > Mn. Grain boundary sliding is associated with continuous breaking and reforming of interlayer chemical bonds, as is the case of defect migration. Therefore, a smaller magnitude of spatial variation of binding strength means a smaller energy change (easiness) in the sliding process. In this scenario, we can understand to some extent the ordering of the beneficial effect of alloying elements on the ductility of Mg alloys, Y < Zn < Mn [38] and the interesting discovery that fine-grained Mg shows large elongation of more than 100%, owing to the activation of grain boundary sliding [39,40,41].

To understand the origin of the remarkable difference demonstrated by Y, Zn, and Mn when they bond with Mg at the twin boundary in Mg, we now compare the density of states (DOS) of these systems. Figure 3 displays the partial DOS for each system. Note that the Mn–Mg bonding is not strong enough to suppress the spin-split of Mn *d* electrons, and it remains spin-polarized in Mg. Since the spin-polarization of *s* and *p* electrons is negligibly small, we display only the summation of up and down spins for them. Different from Y and Mn, the close-shell d states of Zn locate below the *s* and *p* states of Mg, and do not contribute to the Zn–Mg chemical bonding. As a result, the *s* and *p* DOS profiles did not change much with the introduction of Zn into the supercell, and charge transfer induced Fermi level shift is expected. By comparison, both Y–4*d* and Mn–3*d* states were seen to hybridize remarkably with Mg. It is worth noting that although the spin-down states of Mn-3*d* have a significant distribution at the Fermi level, their hybridization with the 3*sp* states of Mg is not as conspicuous as Y–4*d* states, due to the smaller spatial extension of 3*d* electrons.

To have a closer look at and a comparison of the details of the DOS at the Fermi level, we replot the *s*- and *p*-DOS of all four systems in the vicinity of the Fermi energy in Figure 4. Since *s* and *p* states are well separated in magnitude, we use the same color for them and assign different colors for each system. In the highlighted region [−0.3 eV, 0.0 eV], it is seen clearly that both Zn and Mn cause a down-shift of *s*- and *p*-DOS, while Y brings about an up-shift. This is in accordance with the observation in Figure 2 than Zn and Mn gain electrons from Mg, while Y loses electrons. A closer examination reveals that for both *s* and *p* states, Mn made the DOS curve smoother, whereas Zn had such an effect only on the *s*-states, to a lesser extent. A smoother DOS curve near the Fermi energy means a more uniform distribution of the valence electrons in the interstitial region, as shown in Figure 2b. Obviously, the more uniform the charge density in the interlayer region at the grain boundary, the easier the grain sliding.

To further elucidate the effect of spin-polarization on the behavior of Mn, we did a comparative study with non-magnetic treatment of Mn. It was found that in the spin-restricted case, the charge density redistribution is more significant (Figure 5 and Table 2), and the DOS near the Fermi level is more seriously altered (Figure 6). A rougher distribution of the bonding charges in the interlayer interstitial region implies that non-magnetic Mn will not promote grain boundary sliding as the magnetic Mn does. This result clearly shows that “magnetism” is one of the important characteristics in selecting the alloying element to enhance grain boundary plasticity.

We note that in a recent study on the effect of alloying elements on flow stress obtained tensile tests at 373 K using Mg binary alloy [42], we demonstrated that the flow stress in the Mg–Mn alloy is lower than that in pure Mg, although this alloy is expected to show solid solution hardening associated with dislocation slip. This suggests that the other deformation mechanism, i.e., grain boundary sliding, occurs in the Mg–Mn alloy.

Having realized that the strong spin-polarization and small atomic size made Mn a grain sliding enhancer in Mg, we are curious as to whether other magnetic transition metals show the same behavior. We then performed similar first-principles calculations on the effect of Fe, Co, and Ni on the electronic properties of this twin boundary in Mg. Note that all these three elements have a maximum solubility in Mg less than 0.01% [43]. Similar to Mn and Zn, all of the three elements have a strong tendency to segregate to the ($\bar{2}$114) twin boundary in Mg, with the segregation energy being 1.11, 1.03, and 1.08 eV, respectively. However, only Fe remains to be spin-polarized when added into Mg. The spin-split of Fe (Figure 6) is smaller than that of Mn and the *d*-DOS near the Fermi energy is higher than the latter system. Both Co and Ni make the *s*- and *p*-DOS at the Fermi energy much more oscillative than Mn and Fe do. Therefore, we predict that none of the three elements will be effective in enhancing the grain boundary sliding, if they have the characteristic for solving (alloying) into Mg.

## 4. Conclusions

Although both Mn and Y formed strong chemical bonds when added into Mg, Mn was found to enhance grain boundary sliding in Mg, and thus promote the ductility of the alloy, whereas Y is known to suppress grain boundary sliding. In an attempt to reveal the physical origin behind this disparity, we carried out a first-principles density functional theory investigation on the segregation and chemical bonding of Y, Zn, and Mn at the ($\bar{2}$114) twin boundary in Mg. Our calculations demonstrated that all the three elements tend to segregate to the twin boundary. Y preferred a substitutional position near the boundary; Zn and Mn, on the other hand, occupied interstitial positions in the boundary layer due to their small atomic size. The positioning of *d* states of the alloying element with respect of the Fermi energy of the Mg alloy played a vital role in the distribution of the bonding charges. Both Y–4*d* and Mn–3*d* states hybridized with the Mg–3*sp* states, while Zn–3*d* states fell below the energy range of Mg–3*sp* states. As a result, Zn–Mg bonding was characterized by charge transfer only. Different from the Y–4*d* states, which located near the Fermi energy, strong spin-polarization of Mn pushed the up-spin 3*d* states down, leading to less anisotropic Mn–Mg bonds with more delocalized charge distribution in the interlayer region near the twin boundary, and hence promoted grain boundary sliding. This argument was confirmed with alloying addition of spin-restricted Mn, spin-unrestricted Fe, Co, and Ni under the same calculation setting. Spin-restriction of Mn increased remarkably the localization of bonding charges due to the up-shift of Mn–3d orbitals. Moreover, Fe, Co, and Ni were all detrimental in promoting grain boundary sliding and elongation-to-failure.

## Figures and Tables

**Figure 1 materials-15-03483-f001:**
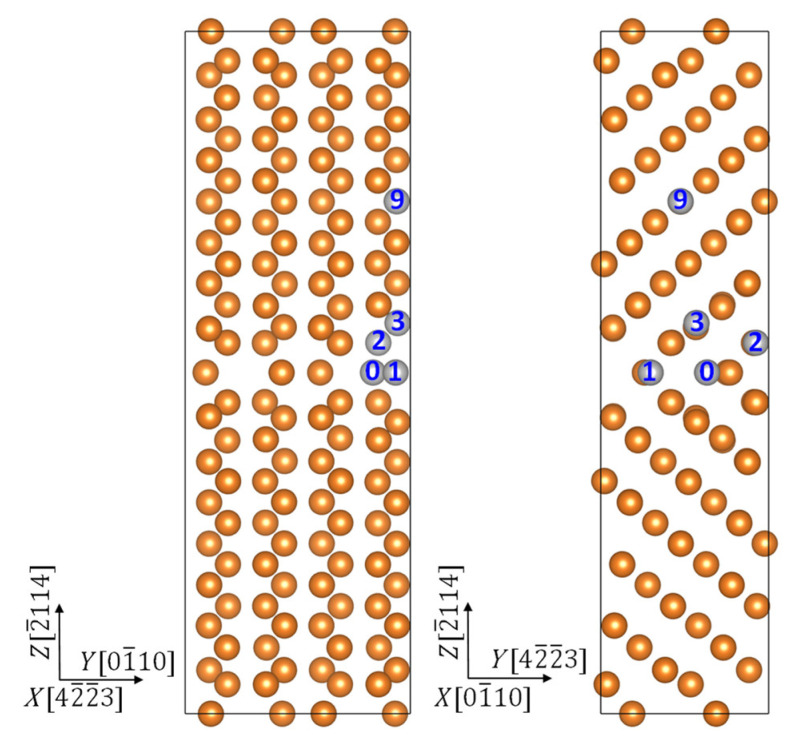
Side views of the supercell used to model the twin boundary segregated with alloying elements. Label “0” denotes the interstitial position at the boundary, and “1”, “2”, “3”, and “9” denote the substitutional positions near the boundary and inside the grain.

**Figure 2 materials-15-03483-f002:**
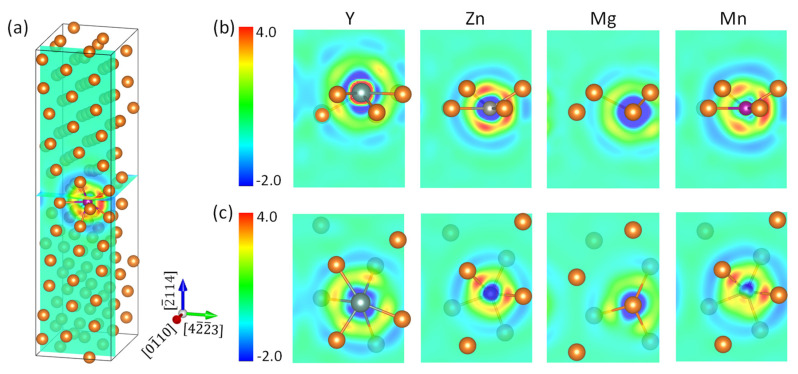
The charge redistribution (*me*/Å^3^) upon formation of the X–Mg bond (X = Y, Zn, Mg, and Mn) at the $\bar{2}$114) twin boundary in Mg. (**a**) Positions of the (0$\bar{1}$10) plane containing element X and the ($\bar{2}$114) plane in between GB1 and GB2 in the supercell. (**b**,**c**) The charge redistribution in (0$\bar{1}$10) and ($\bar{2}$114) planes, respectively.

**Figure 3 materials-15-03483-f003:**
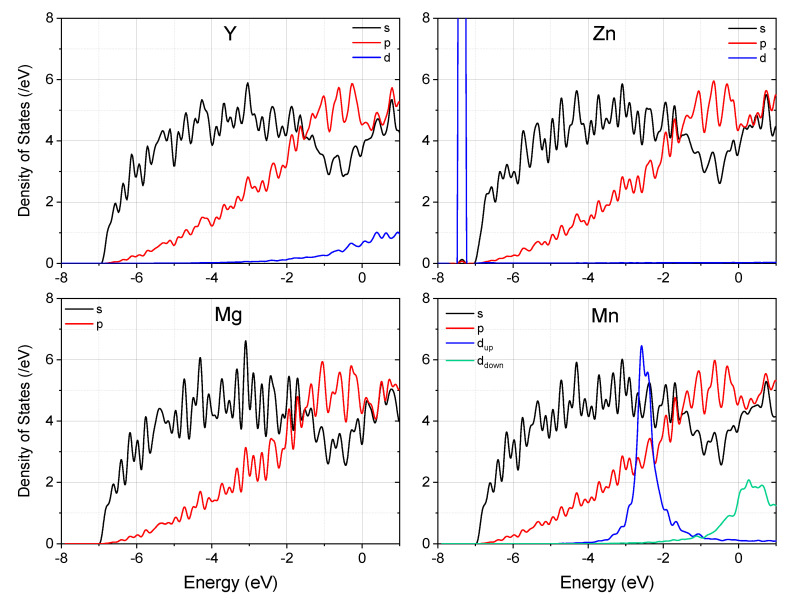
The partial density of states (PDOS) of the supercell with and without segregation of alloying elements at the ($\bar{2}$114) twin boundary in Mg. Note that the spin-up and spin-down *d* states of Mn are colored differently. The Fermi energy is set to zero.

**Figure 4 materials-15-03483-f004:**
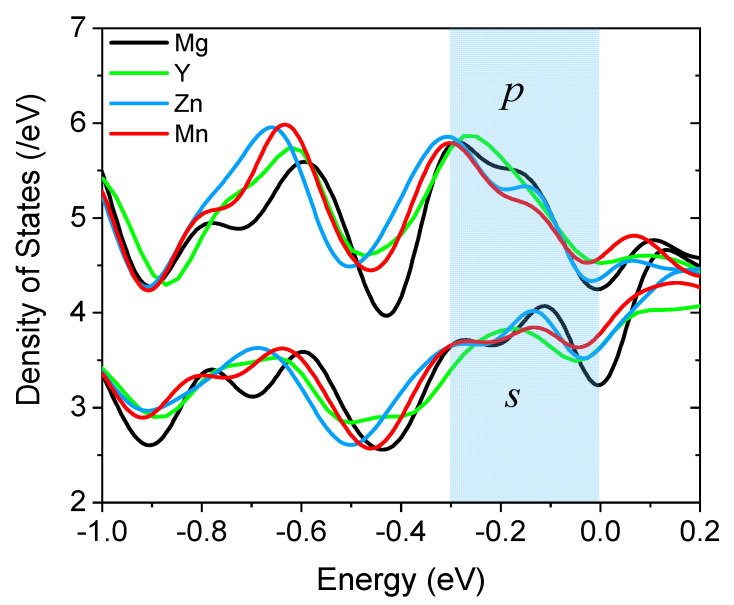
The total density of *s*- and *p*-states near the Fermi energy (set to zero) of the supercell with and without segregated alloying elements.

**Figure 5 materials-15-03483-f005:**
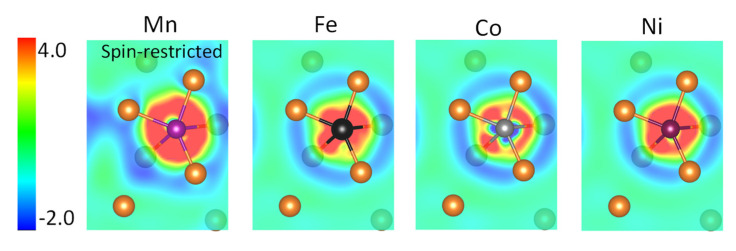
The charge redistribution (meV/Å^3^) in the ($\bar{2}$114) plane upon formation of Mg (A = Fe, Co, and Ni, and non-magnetic Mn) bond at the ($\bar{2}$ 114) twin boundary in Mg. Mn is spin restricted.

**Figure 6 materials-15-03483-f006:**
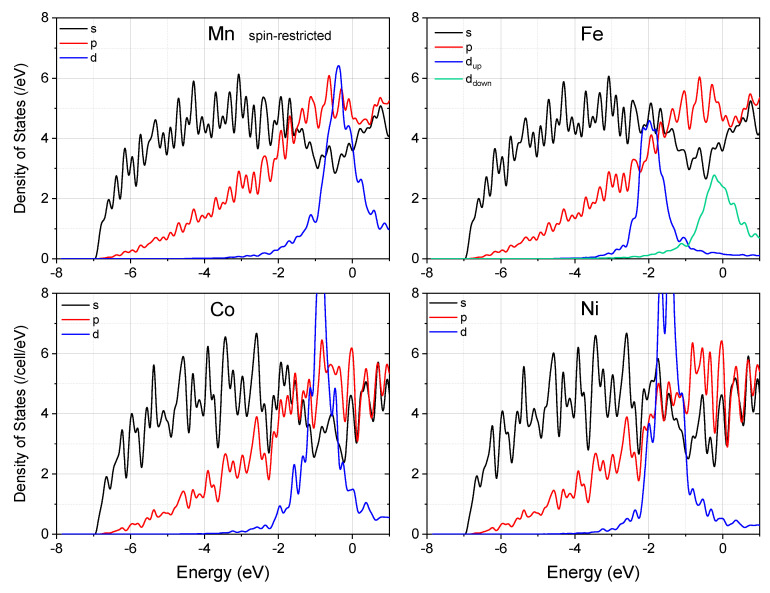
The partial density of states of the supercell with segregation of alloying elements at the ($\bar{2}$114) twin boundary in Mg. Note that the spin-up and spin-down *d* states of Fe are colored differently. Fermi energy is set to zero.

**Table 1 materials-15-03483-t001:** The variation of charge density redistribution (*e*/Å^3^) upon formation of X–Mg (X = Y, Zn, Mg, and Mn) bonds in the (0$\bar{1}$ 10) and ($\bar{2}$ 114) planes at the ($\bar{2}$ 114) twin boundary in Mg (see Figure 2).

		Y-Mg	Zn-Mg	Mg-Mg	Mn-Mg
(0$\bar{1}$10)	Δρ_min_	−0.141	−0.021	−0.004	−0.002
	Δρ_max_	+0.012	+0.004	+0.003	+0.015
	Δρ_max_-Δρ_min_	0.153	0.025	0.007	0.017
($\bar{2}$114)	Δρ_min_	−0.004	−0.005	−0.004	−0.002
	Δρ_max_	+0.008	+0.004	+0.003	+0.004
	Δρ_max_-Δρ_min_	0.012	0.009	0.007	0.006

**Table 2 materials-15-03483-t002:** The variation of charge density redistribution (*e*/Å^3^) upon formation of X–Mg (X = Fe, Co, Ni, and non-magnetic Mn) bonds in the ($\bar{2}$ 114) plane at the ($\bar{2}$ 114) twin boundary in Mg (see Figure 5).

	Mn-Mg Non-Magnetic	Fe-Mg	Co-Mg	Ni-Mg
Δρ_min_ Δρ_max_ Δρ_max_-Δρ_min_	−0.012	−0.014	−0.010	−0.023
	+0.010	+0.019	+0.051	+0.008
	0.022	0.033	0.061	0.031

## Data Availability

All the data is available within the manuscript.

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
