# Peer review of "Spin Polarization of Mn Could Enhance Grain Boundary Sliding in Mg"

_materials, 2022, doi:10.3390/ma15103483_

Round 1
Reviewer 1 Report
“Spin polarization of Mn could enhance grain boundary sliding in Mg” is theoretical research work. Research design is suitable. Computation is well described. Results and discussion are presented well, leading to supportive conclusions.
I endorse to accept the submitted manuscript in its present form.
Author Response
Thank you so much for your appreciation of our research work. We are strongly encouraged to advance our understanding of the effect of spin-polarization in the deformation mechanism in Mg alloys in more extensive investigations.
Reviewer 2 Report
This manuscript offers a concise yet compelling report on the role that spin polarization of Mn could play in the enhance of grain boundary sliding in Mg. A working model of grain boundary is developed and employed. The chosen model, supercell sizes and not least the chosen level of theory all contribute to the elaboration of a successful and reliable protocol for simulating such phenomena as grain boundary sliding. The segregation and its characteristic, the segregation energy, are emphatically emphasized and presented. The authors also provides an adequate and detailed analysis related to the model used as well as well explained and argued motivation behind such a work. From practical point of view, the reported results bring new knowledge and certainly represent an original contribution in the present context.
Thus, the ambitious task in this work covers an array of development including modeling the role of spin polarization in mechanical properties of materials with definitive perspectives for groundbreaking applications that are currently attracting much research interest.
The authors chose an adequate structure of the manuscript – an excellent point of departure for such a study. Finally, the authors provided a balanced realistic and nicely illustrated presentation of their results and corresponding analysis that is of much scientific and practical interest and adds new knowledge to the field.
In my opinion, the fine detailing in the present work, the insightful and balanced discussion of the results, as well as the very good figures, permit competent readers to utilize the manuscript as a guidance for future work. Consequently, this manuscript presents an efficient and beneficial basis for promoting and solving next step challenges in this field.
Moreover, the manuscript benefits from a clear motivation and it is an easy and informative read. The manuscript is also excellent in terms of clarity and accuracy of language.
The present manuscript is a significant contribution, this work once published would be quite useful as well as instructive and suggestive in terms of further studies and to a wider readership.
There are some minor issues with this already excellent manuscript that will need to be addressed before becoming suitable for publication, i.e., it can be considered for publication after a minor revision:
1: In the introduction, the authors partly miss the employment of spin polarized DFT calculations at the same level as in this work for studying stacking and morphological aspects in novel nanostructured materials , e.g., The Journal of Physical Chemistry C 118 (2014) 5501-5509; Journal of Physics: Condensed Matter 27 (2015) 485306. Such works are supportive to the transferability and applicability of similar level of theory schemes applied as used in the present manuscript.
2: Segregation energy is conceptually central to this manuscript, maybe it will be appropriate to introduce a more explicit definition of it in more concrete terms and discuss the convention behind.
3: The authors do mention temperature and thermal aspects of grain boundary sliding in materials only in the introduction. Then no reference to thermal stability and temperature as a factor can be found in the results and discussion section. Were any studies along the thermal stability line being conducted, are there any comments related to the results (except that DFT results do not account directly for the temperature) to be added?
4: Spell-check and stylistic revision of the paper are still necessary. Some long sentences, misspellings, etc., still are noticeable throughout the text.
Author Response
---- Response:
Thank you so much for your appreciation of our research work.
1: In the introduction, the authors partly miss the employment of spin polarized DFT calculations at the same level as in this work for studying stacking and morphological aspects in novel nanostructured materials, e.g., The Journal of Physical Chemistry C 118 (2014) 5501-5509; Journal of Physics: Condensed Matter 27 (2015) 485306. Such works are supportive to the transferability and applicability of similar level of theory schemes applied as used in the present manuscript.
---- Response:
At the end of the introduction, we have added two references and a short statement as: Since spin polarization has been found to play key roles in stacking and morphology in novel nanostructured materials [JPCC 118 (2014) 5501-5509; JPCM 27 (2015) 485306], we have paid particular attention to this effect in this study.
2: Segregation energy is conceptually central to this manuscript, maybe it will be appropriate to introduce a more explicit definition of it in more concrete terms and discuss the convention behind.
---- Response:
In page 3, we have added equation (3) to provide a more explicit definition of segregation energy,
(3)
3: The authors do mention temperature and thermal aspects of grain boundary sliding in materials only in the introduction. Then no reference to thermal stability and temperature as a factor can be found in the results and discussion section. Were any studies along the thermal stability line being conducted, are there any comments related to the results (except that DFT results do not account directly for the temperature) to be added?
---- Response:
In page 9, we have added a short discussion to address this issue, which read as,
We note that in a recent study on the effect of alloying elements on flow stress obtained tensile tests at 373 K using Mg binary alloy [Metal Mater Trans 48A (2017) pp.1366-1374], we have demonstrated that the flow stress in Mg-Mn alloy is lower than that in pure Mg, although this alloy is expected to show solid solution hardening associated with dislocation slip. This suggests that the other deformation mechanism, i.e., grain boundary sliding, occurs in the Mg-Mn alloy.
4: Spell-check and stylistic revision of the paper are still necessary. Some long sentences, misspellings, etc., still are noticeable throughout the text.
----Response
Thanks, we have checked and made corrections in revising the manuscript.
Reviewer 3 Report
Revision of “Spin polarization of Mn could enhance grain boundary sliding 2 in Mg”
The manuscript under review devoted to performed a first-principles comparative study on the segregation and chemical bonding of Y, Zn, and Mn at the Mg (-2114) twin boundary. Providing of such investigations is very useful due to Mg is lightest and its alloys the most important and interesting metallic materials in both industry and fundamental research.
It was found that enhance grain boundary sliding in Mg and thus promote the ductility of the alloy whereas Y is known to suppress grain boundary sliding, although both Mn and Y form strong chemical bonds when added into Mg. To reveal the physical origin behind this disparity first-principles density functional theory investigation on the segregation and chemical bonding of Y, Zn, and Mn at the (-2114) twin boundary in Mg was performed. Calculations demonstrate that all the three elements tend to segregate to the twin boundary. Y prefers a substitutional position near the boundary; Zn and Mn, on the other hand, will occupy interstitial positions in the boundary layer due to their small atomic size. The positioning of d states of the alloying element with respect of the Fermi energy of the Mg alloy plays a vital role in the distribution of the bonding charges. Both Y-4d and Mn-3d states hybridize with the Mg-3sp states, while Zn-3d states fall below the energy range of Mg-3sp states. It was revealed that Zn-Mg bonding is characterized by charge transfer only. Different from the Y-4d states which locate near the Fermi energy, strong spin-polarization of Mn pushes the up-spin 3d states down, leading to less anisotropic Mn-Mg bonds with more delocalized charge distribution in the interlayer region near the twin boundary, and hence promotes grain boundary sliding. The authors point out that this argument is confirmed with alloying addition of spin-restricted Mn, spin-unrestricted Fe, Co, and Ni under the same calculation setting. Spin-restriction of Mn increase remarkably the localization of bonding charges due to the up-shift of Mn-3d orbitals. Moreover, Fe, Co, and Ni are all detrimental in promoting grain boundary sliding and elongation-to-failure.
In manuscript all necessary information is captured by 5 figures, 2 tables and 40 references, all of them are adequate and are reflected in the text.
The obtained results are important both for understanding the physical processes that occur in real objects and for the development of new materials. It corresponds to the field of the Journal «Materials» and may accepted.
Author Response
Thank you so much for your appreciation of our research work. We are strongly encouraged to study the effect of spin-polarization in the deformation mechanism in more extensive alloys.